# QTL Mapping and Prediction of Haploid Male Fertility Traits in Maize (*Zea mays* L.)

**DOI:** 10.3390/plants9070836

**Published:** 2020-07-03

**Authors:** Yanyan Jiao, Jinlong Li, Wei Li, Ming Chen, Mengran Li, Wenxin Liu, Chenxu Liu, Shaojiang Chen

**Affiliations:** National Maize Improvement Center of China, College of Agronomy and Biotechnology, China Agricultural University, Yuanmingyuan West Road, Haidian District, Beijing 100193, China; jiaoyanyan@cau.edu.cn (Y.J.); lijinlong2017@cau.edu.cn (J.L.); wellion@cau.edu.cn (W.L.); acm2638@163.com (M.C.); lmran1996@163.com (M.L.); wenxinliu@cau.edu.cn (W.L.)

**Keywords:** QTL, haploid male fertility (HMF), anther emergence, genomic prediction, pollen production, maize

## Abstract

Chromosome doubling of maize haploids is a bottleneck in the large-scale application of doubled haploid (DH) technology. Spontaneous chromosome doubling (SCD) of haploid has been taken as an important method in the production of DH lines and low haploid male fertility (HMF) is a main limiting factor for the use of SCD. To study its genetic basis, haploids of 119 DH lines derived from a cross between inbred lines Qi319 and Chang7-2 were used to map the quantitative trait locus (QTL) contributing to HMF. Three traits including anther emergence rate (AER), anther emergence score (AES) and pollen production score (PPS) of the haploid population were evaluated at two locations. The heritability of the three traits ranged from 0.70 to 0.81. The QTL contributing to AER, AES and PPS were identified on the chromosomes 1, 2, 3, 4, 5, 7, 9 and 10. Five major QTL, *qAER5-1*, *qAER5-2*, *qAES3*, *qPPS1* and *qPPS5*, were found and each could explain more than 15% of the phenotypic variance at least in one environment. Two major QTL, q*PPS1* and *qPPS5*, and two minor QTL, *qAES2* and *qAER3,* were repeatedly detected at both locations. To increase the application efficiency of HMF in breeding programs, genomic prediction for the three traits were carried out with ridge regression best linear unbiased prediction (rrBLUP) and rrBLUP adding QTL effects (rrBLUP-QTL). The prediction accuracies of rrBLUP-QTL were significantly higher than that by rrBLUP for three traits (*p* < 0.001), which indirectly indicates these QTL were effective. The prediction accuracies for PPS were 0.604 (rrBLUP) and 0.703 (rrBLUP-QTL) across both locations, which were higher than that of AER and AES. Overall, this study provides important information to understand the genetic architecture of SCD of maize haploids.

## 1. Introduction

Doubled haploid (DH) technology can increase breeding efficiency in maize by shortening breeding cycles and saving expenses as compared with conventional methods [1,2] and, therefore, has become a key modern breeding technology [3]. DH technology of maize includes haploid induction, identification and chromosome doubling, DH evaluation and application [3,4]. Over the past 20 years, haploid production has become more and more effective, owing to the high efficiency of in vivo haploid inducers. Some valid methods for haploid identification, such as the *R1-nj* marker [5,6], oil content [7,8], near infrared spectroscopy [9] and hyperspectral imaging [10], have been reported and applied in maize breeding. In contrast, haploid genome doubling still represents a bottleneck of DH technology for large-scale applications.

Generally, haploids are sterile, while fertile gametes could be produced when the chromosomes are doubled. Most of the current chromosome doubling methods depend on antimitotic chemical reagents, such as colchicine, amiprophos-methyl (APM) and pronamide [11,12,13,14]. Traditionally, colchicine treatment of haploids results in a better doubling effect. These chemical reagents are toxic and require special treatments [15,16,17]. Nitrous oxide (N_2_O) is a relatively safe gas that shows some chromosome doubling effects [18]. The treatment of haploid seedlings with N_2_O leads to a similar chromosome doubling rate as does herbicide treatment [19]. However, this method is tedious and specialized equipment is required [20]. Therefore, simple and safe chromosome doubling methods are needed.

By comparison, spontaneous chromosomal doubling (SCD) is considered an important method without chemical treatment. The fertility of haploids includes male and female fertility [21], with the rate of haploid female fertility being higher than that of haploid male fertility (HMF) [22,23]. Therefore, low HMF is regarded as the main limiting factor in the production DH lines by way of SCD. A previous study noted that the HMF rate of 20 elite inbred lines ranged from 9.8% to 89.8% [24], and the HMF rate of temperate maize germplasms was much higher than that of tropical germplasms [25]. A significant difference for the haploid pollen production score (PPS) was observed between locations [24], and the proportion of fertile haploids differed between field conditions and greenhouse conditions, ranging from as high as 20% in the field to 70% under greenhouse conditions [25]. Therefore, HMF is affected by the genotype and by the environment. Recent studies revealed substantial genetic variation and high heritability of HMF [1,24,26], and thus materials with a high HMF rate should be used for genetic studies.

Several studies have been carried out to understand the genetic basis of SCD. Quantitative trait locus (QTL) controlling HMF have been identified on 10 maize chromosomes in previous research, Ren et al. detected four QTL controlling HMF and fine-mapped *qhmf4*, a key QTL on chromosome 6 [27]. Ma et al. used genome-wide association mapping to identify 4 HMF QTL in the Zheng58 background and 16 HMF QTL in the Mo17 background, these QTL jointly explain 22.5% of the phenotypic variance [26]. Chaikam et al. detected eight significant single-nucleotide polymorphisms (SNPs), which in total explain 34% of the phenotypic variance for HMF [28]. Yang et al. also identified nine QTL in F2:3 families from the cross of Zheng58 and K22 [29]. Although some HMF QTL have been reported, their contributions to HMF traits are small. Thus, further studies on the genetic architecture of HMF and genomic based prediction are still required for highly efficient application of SCD in DH breeding.

In this study, haploids from 119 DH lines derived from a cross between maize inbred lines Qi319 and Chang7-2 were used to detect QTL controlling HMF-related traits, anther emergence rate (AER), anther emergence score (AES) and PPS in different environments. The objectives of this study were to (1) investigate the phenotypic variation of HMF-related traits, (2) identify the QTL associated with HMF in maize and find novel and stable loci across different environments and (3) evaluate HMF prediction models. The resulting QTL and prediction model could be used for improving the HMF and facilitating application of DH technology.

## 2. Results

### 2.1. Phenotypic Evaluation of HMF Traits for DH Population

HMF of DH-derived haploids and haploids of parental lines were evaluated at two locations (Figure 1), phenotypic evaluation revealed that HMF-related traits of Chang7-2 haploids were higher than those of Qi319 haploids (Figure 1A,B). The AER of Chang7-2 and Qi319 haploids was <20%, AES and PPS of Chang7-2 and Qi319 haploids were <0.2 in Beijing; whereas the AER, AES and PPS of Chang7-2 haploids in Hainan were 65.98%, 0.20 and 0.52, and 44.04%, 0.14 and 0.33 across both locations, respectively (Figure 1B). More than 50% DH lines could produce haploids with HMF traits exceeds that of haploids derived from parental lines (Figure 1A). Thus, we assumed that this population could be further used to map the QTL contributing to HMF. Mean values of AER, AES and PPS for haploid population were 35.56%, 0.15 and 0.25 in Beijing, 72.48%, 0.36 and 0.55 in Hainan and 53.07%, 0.26 and 0.39 across both locations, respectively (Figure 1B). The phenotype for each of the three traits among the DH-derived haploid population grown in Beijing presented a typical skewed distribution (Figure 1A), with a vast majority of haploids having low score levels of HMF-related traits. In contrast, more haploids had high score levels of AER in Hainan. AES and PPS also showed a skewed distribution among the Hainan-grown population, but the coefficients of skew were smaller than that in Beijing. Taken a score above 70% for AER or 0.7 for AES and PPS as high levels HMF, the percentages of high level male fertility haploids were 23.20%, 1.60% and 12.00% across both locations for AER, AES and PPS, respectively (Figure 1C). Variation statistics revealed that AER, AES and PPS were significantly different (*p* < 0.01) with respect to genotype, location and the interaction of genotype by location for the DH-derived haploid population (Figure 1D). The heritability (*h^2^*) of AES was 0.81, higher than that of PPS (0.79) and AER (0.70) (Figure 1D). The above results suggested that HMF traits are mainly determined by genotype but are also affected by the environment.

### 2.2. QTL Analyses of Three HMF-Related Traits

Considering the effects of environment on HMF-related traits, we carried out QTL mapping with the phenotypic data collected in Beijing, Hainan and the best linear unbiased prediction values for two locations. A total of eight QTL for AER, seven QTL for AES and eight QTL for PPS were detected on chromosomes 1, 2, 3, 4, 5, 7, 9 and 10, including four QTL, *qAER3, qAES2 qPPS1* and *qPPS5* were repeatedly detected (Figure 2, Table 1, Table 2 and Table 3).

#### 2.2.1. QTL for AER

The phenotypic contributions of individual QTL for AER ranged from 0.21 to 19.69% (Table 1). *qAER5-1* was identified with a phenotypic variance explanation (PVE) of 14.60% and 19.51%, respectively, in Beijing and across both locations, and *qAER5-2* was identified with a PVE of 19.69% in Hainan. Both QTL were located on chromosome 5 and the favorable alleles came from Chang7-2. In addition, the minor QTL *qAER3* on chromosome 3 had a PVE of 8.02%, 6.41% and 7.80% in Beijing, Hainan and across both locations, respectively (Table 1), and Qi319 was the source of the AER-increasing alleles (Table 1).

#### 2.2.2. QTL for AES

Seven QTL for AES were identified, including four repeatedly detected QTL (Table 2), *qAES2* on chromosome 2 was identified three times, and *qAES1* on chromosome 1 was detected twice. These two QTL do, however, represent minor QTL, as their PVE values were <10% (for two locations), and Qi319 was the source of the AES-increasing alleles in both cases (Table 2). Two major QTL, designated *qAES3* and *qAES9*, were detected in Hainan and across both locations, and they explained 15.61% and 10.28% of phenotypic variation in Hainan and 10.59% and 8.4% of phenotypic variation across both locations, respectively (Table 2).

#### 2.2.3. QTL for PPS

All QTL for PPS were repeatedly detected at least twice except *qPPS3-1* and *qPPS2* (Table 3). *qPPS1* and *qPPS5* were two major QTL with favorable alleles from the inbred line Chang7-2, which explained more than 15% of phenotypic variance at least once (Table 3). In addition, four minor QTL, *qPPS3-2*, *qPPS7*, *qPPS9* and *qPPS10*, were repeatedly detected twice and explained 4.49%, 8.48%, 6.46% and 7.70% of phenotypic variation in Hainan and 3.70%, 6.26%, 4.22% and 9.12% of phenotypic variation across both locations, respectively (Table 3). Furthermore, *qPPS3-2* maybe overlapped with *qAER3* and *qAES3*. For the other two QTL, *qPPS2* and *qPPS3-1*, were identified only once, and the PPS-increasing allele originated from Qi319 (Table 3).

### 2.3. The Segregation Distortion of the Putative QTL for Three HMF-Related Traits

The segregation ratio for each putative QTL was tested by χ^2^ test, and the 14 QTL showed segregation distortion. Four AER-QTL, *qAER1*, *qAER3*, *qAER5-1* and *qAER5-2*, showed segregation distortion at least once (Table 1). Among AES-related QTL, only *qAES4* and *qAES5-2* fitted the expected Mendelian segregation ratio of 1:1; the other AES QTL showed segregation distortion in at least once (Table 2). All PPS-QTL showed segregation distortion in at least once except those for *qPPS3-1, qPPS3-2* and *qPPS9* (Table 3). Thus, most QTL, including eight major QTL, showed segregation distortion near their peak markers. Segregation distortion of QTL were found to be biased to the corresponding donor parent (Table 1, Table 2 and Table 3).

### 2.4. Genomic Prediction (GP) for Three HMF-Related Traits

GP for HMF of haploids would be an alternative efficient way to improve DH breeding schedule and efficiency. GP for three HMF-related traits was implemented by ridge regression best linear unbiased prediction (rrBLUP) and rrBLUP adding QTL effects (rrBLUP-QTL) models. Prediction accuracies for the three HMF-related traits ranged from 0.274 to 0.703 (Figure 3). Compared with the prediction accuracies among these traits and locations, the PPS across two location had the highest prediction accuracy by either rrBLUP (0.640) or rrBLUP-QTL (0.703) model. The prediction accuracy of rrBLUP-QTL was significantly higher than that by rrBLUP for three traits (*p* < 0.001). For the three HMF-related traits, the prediction accuracy of PPS was higher than that of AER and AES, and reached medium levels (Figure 3). The prediction accuracies of HMF traits at Hainan were higher than at Beijing, except for AER.

## 3. Discussion

### 3.1. The Evaluation Standard of HMF

Chromosome doubling of both male and female gametophytes are necessary for achieving the fertility of haploids. In comparison with of female gametophytes, production of fertile male gametophytes is more difficult, and therefore, the deep understanding of the genetic basis for HMF traits is necessary to increase the efficiency of SCD in DH breeding. The emergence of anthers and shedding of pollen seem two important but independent processes that occur before pollination. For example, in our preliminary experiments, inbred line Yu87-1 shows a high anther emergence (>75% of anthers emerged on tassels); however, few anthers could shed pollen (data not shown). In contrast, the other inbred line, Chang7-2, showed low anther emergence, but the pollen was easily shed from the anthers, even though just a few anthers were visible on each tassel (data not shown). To accurately assess HMF, both anther emergence and pollen production should be considered. In previous genetic analyses of HMF, only anther emergence, either AES or AER was considered and used as an evaluation standard for HMF [24,26,27,28,29,30]. In a study by Wu et al., four HMF-related traits, AER, AES, PPS and pollen production rate, were used as evaluation indexes to explain the genetic structure of HMF [24]. In this study, three HMF-related traits (AER, AES and PPS) were significant positive correlated (*p* < 0.001), and pairwise correlation coefficients ranged from 0.73 to 0.88 (Table 4), but they each are related to different aspects of HMF. AER is a measure of the percent of haploid plants with anther emergence in a haploid population; however, AES and PPS are measures of the male fertility of each haploid. It may provide more information when all of the three traits are considered in QTL mapping of HMF traits, which would be beneficial for providing a deep understanding of anther emergence, pollen shedding and, finally, the HMF of haploids.

### 3.2. Factors Affecting HMF

We observed the three HMF related traits showed significantly different (*p* < 0.01) between the two locations and performed the interaction of genotype by environment (Figure 1C), this result was consistent with the previous findings [24,25]. The spontaneous chromosome doubling performance of haploids grown in Hainan was higher than that in Beijing (Figure 1A). A similar phenomenon for HMF was observed in a similar study between the haploids grown in Hainan and Zhengzhou [29]. These differences may be attributed to temperature regimes, photoperiod or other environmental factors. All three HMF-related traits presented a typical skewed distribution among Beijing-grown plants (Figure 1A), where most of the haploids showed low HMF levels. In contrast, for plants grown in Hainan, the proportion of plants with high AER values was larger than that with low values. AES and PPS also showed a skewed distribution among the Hainan-grown plants, but the coefficients of skew were smaller. The effect of the environment on AER was thus greater than that on PPS and AES. Our data suggest that haploids could be planted at a location that facilitates the spontaneous recovery of haploids, such as the Hainan winter nursery, which would be beneficial for increasing HMF and producing more DH lines.

### 3.3. QTL for HMF

In this study, the *h*^2^ of three HMF-related traits ranged from 0.70 to 0.81 (Figure 1D), which is similar to estimates calculated in previous studies [24,25]. Therefore, it is applicable to identify QTL controlling HMF using this population. Furthermore, 23 QTL located on chromosomes 1, 2, 3, 4, 5, 7, 9 and 10 were identified in this DH population. Among the QTL identified here, the major QTL located in bins 3.04/05 and 5.03/04 and three minor QTL located in bins 2.05, 4.05 and 7.02 correspond to previously isolated QTL [26,27,29,30], and the major QTL located in bins 1.05 and 9.03 and the minor QTL located in bins 10.06/10.07 were newly identified QTL that had different positions than those identified in previous study [26,27]. In addition, the four QTL on chromosome 3, *qAER3*, *qAES3*, *qPPS3-1* and *qPPS3-2*, may overlap, as their positions are close to one another, and may thus control AER, AES and PPS simultaneously. Five QTL located on chromosome 5 were detected among the population in two locations, among which *qAER5-1*, *qAES5-1*, *qAES5-2* and *qPPS5* perhaps represent the same locus because of their similar positions and contributions to AER, AES and PPS. However, whether *qAER5-2* represents a single locus with the above QTL (*qAER5-1*, *qAES5-1*, *qAES5-2* and *qPPS5*) will require further analysis. Further fine-mapping of these QTL will provide valuable information to benefit the development of high-HMF breeding materials and to clone their underlying genes.

The segregation ratio of most QTL for HMF-related traits showed significant segregation distortion in this study (Table 1, Table 2 and Table 3).The QTL mapping based on segregation distortion has been verified with plant height in a haploid population, and was further used to identify four QTL controlling HMF [27]. A haploid population, including many individuals with higher level HMF as score of ≥70% for AER or ≥0.7 for AES and PPS, was used to analyze segregation distortion of the most significant marker in this study. Favorable alleles for HMF could be fixed during the production of DH lines, haploids with those favorable alleles would be easier to produce fertile pollen than those without. As false-positive QTL detected by CIM could be examined again by analyzing the genotype distortion in this population with an extreme phenotype, these segregation distortion-related results provide a reference for future studies.

### 3.4. GP for HMF Application in Breeding Schedule

The QTL mapping studies up to now have showed that the genetic basis of SCD is complex, and most detected QTL were minor effect QTL [26], thus, GP would be an alternative option to increase efficiency of SCD in practical breeding via selecting genomic estimated breeding values. rrBLUP is widely used to predict genotypic values in breeding practice [31,32], thus, rrBLUP and rrBLUP-QTL were used in this study. Prediction accuracies for the three HMF-related traits analyzed here ranged from 0.274 to 0.703, which is consistent with previous studies [26,28] and suggested that it would be possible to predict HMF performances at different genetic backgrounds. The two models had better prediction ability for PPS, which may be due to its highly heritability. In addition, the prediction accuracies of rrBLUP-QTL were significantly higher than that by rrBLUP for three traits (*p* < 0.001), which indirectly indicated that the QTL information was useful in GP. With the development of genomics, using GP to predict agronomic traits of a haploid would be an important procedure during DH breeding. Thus, it would be valuable to incorporate the HMF prediction in breeding practices, the haploids predicted to have optimum agronomic traits with high spontaneous doubling rates would be selected to produce DH lines directly at low cost, which would increase the efficiency of DH breeding.

## 4. Materials and Methods

### 4.1. Plant Materials and Haploid Induction

A DH population consisting of 119 lines derived from a F_1_ between inbred lines Qi319 and Chang 7-2 [33] was used in this study. Qi319 and Chang7-2 were two elite inbred lines in China, developed by Maize research institute, Shandong academy of agricultural science and Anyang agricultural science institute, respectively. They are the important representative elite lines of the different heterotic groups, and are the parents of many large-scale planted hybrids. A high-efficiency haploid inducer line, CAU6, was used to induce haploids. CAU6 is a maternal haploid inducer in B73 background with the *R1-nj* marker, developed by China Agricultural University. All haploids mentioned in this study refers to the maternal haploids. These DH lines were planted at the Hainan (18°21′ N 109°10′ E) winter nursery of China Agricultural University in 2017, and pollinated by CAU6. Haploids were identified with the *R1-nj* marker [6] and used to study HMF-related traits.

### 4.2. Phenotypic Evaluation and Statistical Analysis

The experiment was conducted at two locations, Shang Zhuang Experimental Station of China Agricultural University in Beijing (39°56′ N 116°20′ E) in the summer of 2018 and the Hainan winter nursery in 2018. At each location, the experiment used a completely randomized block design with two replicates. All haploids of the parents and these DH lines were investigated daily for HMF from the beginning to the end of pollen shedding, and the results from the pollination peak period were used for analysis.

HMF was evaluated by anther emergence and pollen production. Three HMF-related traits, AER, AES and PPS, were used to determine HMF as described [24]. The three traits were calculated as follows (Equations (1)–(4)):(1)$$\mathrm{AER}=\frac{n_{A}}{n}\times 100\%$$
where n_A_ is the total number of haploid plants with emerged anthers and n is the total number of haploid plants per plot.
(2)$$\mathrm{AES}=\frac{\sum^{}\left(n_{\mathrm{Ai}}\times \mu_{\mathrm{Aj}}\right)}{n\times h_{\mathrm{Amax}}}$$
where AES is the weighted mean of μ_Aj_; n_Ai_ is the number of haploids for each level of anther emergence; μ_Aj_ is the AES of each haploid plant, which is classified into grades 1-5; h_Amax_ is the highest level of anther emergence, which is 5; and n is the total number of haploid plants per plot.
(3)$$\mathrm{PPS}=\frac{\sum^{}\left(n_{\mathrm{Pi}}\times \mu_{\mathrm{Pj}}\right)}{n\times h_{\mathrm{Pmax}}}$$
where PPS is the weighted mean of μ_Pj_; n_Pi_ is the number of haploids for each level of pollen production; μ_Pj_ is the PPS of each haploid plant, which is classified into grades 1-5; h_Pmax_ is the highest level of pollen production, which is 5; and n is the total number of haploid plants per plot.

Variance components were estimated by SAS software with the PROC MIXED procedure of REML, and heritability was calculated as follows:(4)$$h^{2}=\frac{\sigma_{G}^{2}}{\sigma_{G}^{2}+\frac{\sigma_{G\times L}^{2}}{L}+\frac{\sigma_{e}^{2}}{L\times R}}$$
where $\sigma_{G}^{2}$ is the genotypic variance; $\sigma_{G\times L}^{2}$ is the genotype × location interaction variance; $\sigma_{e}^{2}$ is the residual error variance; *L* is the number of plant locations; and *R* is the number of replications, which is 2. This formula was also used for the calculation of h^2^ for AER, AES and PPS.

### 4.3. Genotyping and QTL Mapping

Genotypic data was obtained by SNP chip genotyping, as described in previous study [33]. In this study, the genetic map was constructed with 2036 polymorphic SNPs using Kosambi’s regression function in mapchart2.32 software, in which the logarithm of odds (LOD) value of 2.5 was set as the cut-off for detection of putative QTL. The phenotypic data across both locations represented the best linear unbiased prediction values for two locations and were calculated using R/lmer in which genotype and locations and genotype × locations as random effects. The phenotypic data of Beijing, Hainan and the best linear unbiased prediction values for two locations were used to conduct QTL mapping, QTL with additive effects and dominant effects were analyzed by R/qtl [34] using CIM. QTL that had a peak marker difference of less than 20 cM were one QTL [35]. Of them, QTL that explained more than 10% of phenotypic effects at least once were defined as major-effect QTL here. Taken a score above 70% for AER or >0.7 for AES and PPS as high levels HMF in one DH-derived haploids population, and DH lines that produced high male fertility haploids were selected to analyze segregation distortion for the most significant markers of putative QTL. Whether the allele of the most significant markers fit the expected Mendelian segregation ratio of 1:1 was detected by chisq.test function in R.

### 4.4. GP Analysis

GP for the three HMF-related traits was carried out with two models, rrBLUP [36] and rrBLUP-QTL [31], in the DH mapping population. And the effect of detected-QTL as fixed effects on the rrBLUP-QTL model. To compute accuracies, a five-fold cross-validation scheme was applied and repeated 100 times. For this analysis, the phenotypes associated with Beijing- and Hainan-grown plants and across both locations were used for GP. The prediction accuracy was estimated as the correlation between genomic estimated breeding values by rrBLUP or rrBLUP-QTL and the observed phenotypes.

## 5. Conclusions

In this study, 23 QTL in total controlling HMF-related traits were detected on chromosomes 1, 2, 3, 4, 5, 7, 9 and 10. Among those QTL, 13 were repeatedly detected at least twice, including four QTL, *qAER3*, *qAES2*, *qPPS1* and *qPPS5*, were repeatedly detected three times. In addition, GP for three HMF–related traits demonstrated that the prediction accuracy for PPS was higher than that of AER and AES. These results provide a better understanding for the genetic basis of spontaneous chromosomal doubling of haploid in maize.

## Figures and Tables

**Figure 1 plants-09-00836-f001:**
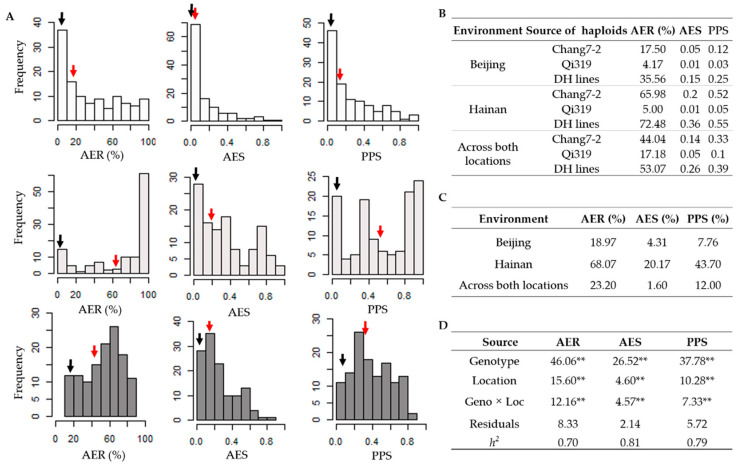
(**A**) The distribution of anther emergence rate (AER), anther emergence score (AES), and pollen production score (PPS) for the haploid mapping population grown in Beijing (white), Hainan (light gray) and across both locations (dark gray). Mean values of haploid male fertility-related traits for Qi319 (black arrows) and Chang7-2 (red arrows) were shown. (**B**) AER, AES and PPS of parental inbred line haploids at different locations. (**C**) The proportion of high male fertility (score of ≥70% for AER or ≥0.7 for AES and PPS) haploids for AER, AES and PPS. (**D**) Variation statistics and heritability (*h^2^*) of AER, AES and PPS in the haploid mapping population. ** *p* < 0.01.

**Figure 2 plants-09-00836-f002:**
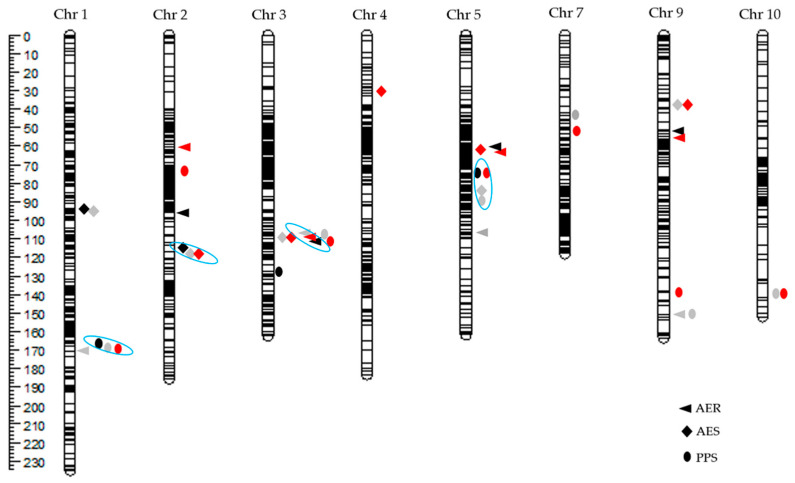
Genetic map and chromosomal locations of quantitative trait locus (QTL) associated with HMF-related traits mapped in the doubled haploid population grown in Beijing (black) and Hainan (gray) and across both locations (red). QTL repeatedly identified were shown in blue circle.

**Figure 3 plants-09-00836-f003:**
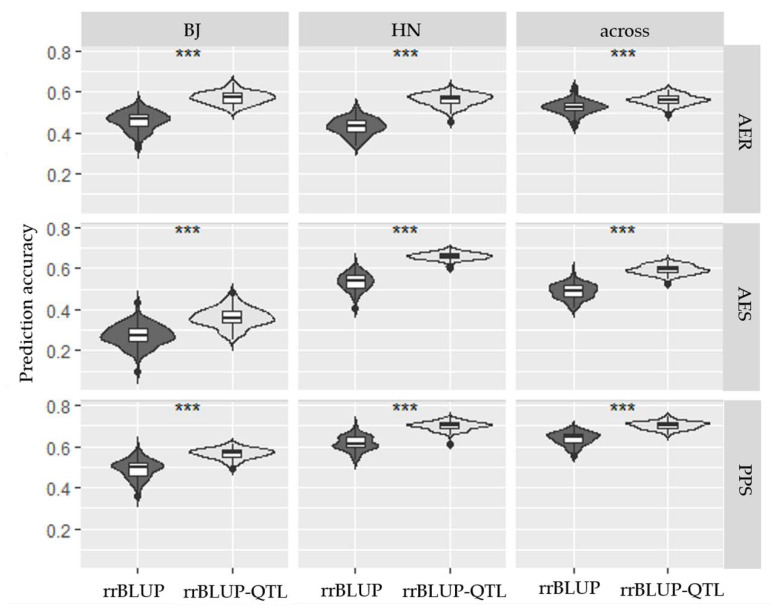
Prediction accuracies for three HMF-related traits generated by rrBLUP and rrBLUP-QTL for Beijing (BJ) and Hainan (HN) and across both locations (across). *** *p* < 0.001, means prediction accuracies as determined by rrBLUP and rrBLUP-QTL were significantly different.

**Table 1 plants-09-00836-t001:** Putative QTL detected for AER in the haploid mapping population.

Environment	QTL Name	Marker	Chr	Position (cM)	LOD	Add	PVE (%)	Allele		χ^2^	*p*-Value
								Q	C		
Beijing	*qAER2-1*	PZE-102130712	2	98.24	2.68	−0.08	0.21	14	8	1.64	0.201
	*qAER3*	PZE-103145047	3	113.47	5.70	−0.26	8.02	17	5	6.55	0.011
	*qAER5-1*	PZE-105077018	5	64.39	3.56	0.27	14.60	2	19	13.76	<0.001
	*qAER9-1*	SYN34182	9	51.12	2.90	0.14	4.72	7	15	2.91	0.088
Hainan	*qAER1*	PZE-101207960	1	173.69	3.49	0.17	4.22	26	55	10.38	0.001
	*qAER3*	PZE-103135808	3	108.06	2.51	−0.17	6.41	46	30	3.37	0.066
	*qAER5-2*	SYN1878	5	109.29	8.21	0.34	19.69	19	62	22.83	<0.001
	*qAER9-2*	PZE-109115897	9	152.41	2.82	0.14	4.00	33	48	2.78	0.096
Across both locations	*qAER2-2*	PZE-102055945	2	62.11	2.83	−0.12	5.13	19	10	2.79	0.095
	*qAER3*	PZE-103138981	3	111.82	4.92	−0.16	7.80	21	8	5.83	0.016
	*qAER5-1*	PZE-105094949	5	67.29	4.71	0.20	19.51	3	26	16.33	<0.001
	*qAER9-1*	PZE-109026940	9	55.90	2.54	0.10	5.64	10	19	2.79	0.095

Marker: most significant marker; Chr: chromosome location; Position: position on chromosome; LOD: logarithm of odds; Add: additive effect (positive and negative values suggest that favorable alleles came from inbred line Chang7-2 and Qi319, respectively); PVE: phenotypic variation explanation; Q: Qi319 allele; C: Chang 7-2 allele. *p* < 0.05 indicates that this marker showed distorted segregation.

**Table 2 plants-09-00836-t002:** Putative QTL detected for AES in the haploid mapping population.

Environment	QTL Name	Marker	Chr	Position (cM)	LOD	Add	PVE(%)	Allele		χ^2^	*p*-Value
								Q	C		
Beijing	*qAES1*	PZE-101098199	1	93.80	2.65	−0.04	1.17	5	0	5.00	0.025
	*qAES2*	PZE-102141171	2	116.96	2.66	−0.11	6.71	4	1	1.80	0.18
Hainan	*qAES1*	PZE-101111793	1	96.70	3.25	−0.14	4.59	19	5	8.17	0.004
	*qAES2*	SYN216	2	120.65	2.71	−0.13	4.55	18	5	7.35	0.007
	*qAES3*	PZE-103138981	3	111.81	6.93	−0.26	15.61	19	5	8.17	0.004
	*qAES5-1*	PZE-105126493	5	86.97	5.16	0.26	11.42	2	22	16.67	<0.001
	*qAES9*	PZE-109019490	9	39.03	4.23	0.18	10.28	4	20	10.67	0.001
Across both locations	*qAES2*	SYN216	2	120.66	4.41	−0.13	6.66	2	0	2.00	0.157
	*qAES3*	PZE-103138981	3	111.82	5.44	−0.13	10.59	1	1	0.00	1.000
	*qAES4*	PZE-104016598	4	31.62	3.70	−0.13	8.60	2	0	2.00	0.157
	*qAES5-2*	PZE-105079012	5	64.36	3.46	0.13	11.29	0	2	2.00	0.157
	*qAES9*	PZE-109019490	9	39.03	3.89	0.11	8.40	0	2	2.00	0.157

**Table 3 plants-09-00836-t003:** Putative QTL detected for PPS in the haploid mapping population.

Environment	QTL Name	Marker	Chr	Position (cM)	LOD	Add	PVE (%)	Allele		χ^2^	*p*-Value
								Q	C		
Beijing	*qPPS1*	PZE-101204728	1	168.12	5.09	0.19	12.66	0	9	9.00	0.003
	*qPPS3-1*	PZE-103162736	3	133.82	2.92	−0.15	8.07	7	2	2.78	0.096
	*qPPS5*	SYN32709	5	77.05	3.34	0.19	9.60	0	9	9.00	0.003
Hainan	*qPPS1*	PZE-101206252	1	170.81	4.95	0.21	12.36	11	41	17.31	<0.001
	*qPPS3-2*	PZE-103135808	3	108.06	3.96	−0.21	4.49	31	18	3.45	0.063
	*qPPS5*	PZE-105136417	5	91.34	6.79	0.34	15.56	5	47	33.92	<0.001
	*qPPS7*	PZE-107110443	7	45.60	3.27	0.21	8.48	10	42	19.69	<0.001
	*qPPS9*	PZE-109115897	9	152.41	3.16	0.16	6.46	21	31	1.92	0.166
	*qPPS10*	PZE-110102769	10	141.81	4.04	−0.19	7.70	35	17	6.23	0.013
Across both locations	*qPPS1*	PZE-101206252	1	170.82	7.22	0.14	16.14	0	15	15.00	<0.001
	*qPPS2*	SYN24086	2	74.71	2.67	−0.09	7.27	12	3	5.40	0.020
	*qPPS3-2*	PUT-163a-13490777-226	3	112.24	5.33	−0.15	3.70	11	4	3.27	0.071
	*qPPS5*	SYN32709	5	77.05	8.96	0.22	14.31	0	15	15.00	<0.001
	*qPPS7*	PZE-107104221	7	53.17	3.17	0.12	6.26	3	12	5.40	0.020
	*qPPS9*	PZE-109112516	9	142.84	2.78	0.08	4.22	5	10	1.67	0.197
	*qPPS10*	PZE-110102769	10	141.81	4.64	−0.13	9.12	11	4	3.27	0.071

**Table 4 plants-09-00836-t004:** Analysis of the relationship between AER, AES and PPS.

Trait	AER	AES	PPS
AER	1	0.80 ***	0.88 ***
AES		1	0.73 ***
PPS			1

*** *p* < 0.001.

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
