# Peer review of "QTL Mapping and Prediction of Haploid Male Fertility Traits in Maize (Zea mays L.)"

_plants, 2020, doi:10.3390/plants9070836_

Round 1
Reviewer 1 Report
The manuscript shows interesting data on the genetic basis of spontaneous chromosome doubling of maize haploids and what’s more important on the detection QTLs controlling haploid male fertility (HMF) related traits, anther emergence rate (AER), anther emergence score (AES) and PPS in two different environments.
The topic is really interesting, even if some modifications are necessary before a reconsideration for publication in Plants. In particular, I have two major concerns about this manuscript: First of all, I suggest to introduce also more detailed data on plant material i.e. please describe this two inbred lines Qi319 and 325 Chang 7-2 (why they were important) in order to evaluate the genetic background of haploid male fertility in obtained 119 DH lines; on the other side, I think that is also necessary to add a clear information about studied environments – are there two or three environments exactly ? This is not clear because in abstract it is written about 3 but in introduction about 2 environments. Please, describe the environmental conditions both in Beijing and Hainan.
These data, I think, are necessary to have a clear idea of the potential of each DH line to be included in future breeding programs.
Moreover, thre is no general conclussions at all.
In the Introduction - line 79: The sentence should begin with the capital letter i.e. " The objective..."
Author Response
Comment 1: Introducing more detailed data on plant material i.e. please describe this two inbred lines Qi319 and 325 Chang 7-2 (why they were important) in order to evaluate the genetic background of haploid male fertility in obtained 119 DH lines
Response: Thank you for your valuable comment and advice.
We selected the population for the following reasons. One is that Qi319 and Chang7-2 are two elite inbred lines for the summer maize area in China, which are the representative elite lines of different heterotic groups, the two lines are the parents of many elite hybrids, such as Qi319 is a female parent of the large scale planted hybrid Ludan981, and Chang7-2 is a male parent of the hybrid Zhengdan958, which is still the top hybrid planted in China, then the two elite lines are very important value in maize breeding. Exploring the spontaneous doubling related QTLs will be easier used for improving the traits in the genetic linked germplasm or the lines. Another reason is that we observed huge phenotypic variance for HMF traits in DH population (Data was shown in Figure 1), which is useful in analyzing genetic basis of HMF traits. The main characteristics and breeder of inbred lines Qi319 and Chang7-2 have been added in line 341-344.
Comment 2: Add a clear information about studied environments – are there two or three environments exactly?
Response: Thank you for your comment. The phenotyping work was carried out at two locations, Beijing and Hainan. In data analyze, the predicted values with phenotypic data from both locations by best linear unbiased method was also analyzed to mapping these QTL. We clarified this part in section 4.3, line 352-353 and 386-392, and corrected three environments to two environments.
An overall conclusion has been added in the end of the manuscript.
Reviewer 2 Report
The manuscript (plants-815706) is targeted to study the genetic basis of spontaneous chromosome doubling of maize haploids. 40 QTLs controlling three traits related to Haploid Male Fertility were detected. Moreover, Genome Selection (GS) for three HMF –related traits was successfully carried out with GBLUP and rrBLUP models in the DH mapping population. This paper is well designed and executed and contains valuable information that must be of interest for readers of genetics and plant breeding. Figures, tables and manuscript structure are clear. The manuscript is suitable for publication in Plants with minor revision.
Author Response
Comment : The manuscript is suitable for publication in Plants with minor revision.
Response: Thank you for your comment. This manuscript has been further revised according to comments from all reviewers, and carefully read by all authors to correct mistakes and typos.
Reviewer 3 Report
General
While the „haploid technology“ is a very useful tool in breeding of maize and other crops, doubling the chromosome complement of maize haploids still is a bottleneck in the practical application of the technology. Besides versious chemical approaches, spontaneous chromosome doubling is an easy and useful method. The present work represents the results of a genetic study on spontaneous chromosome doubling of maize, using 119 DH lines derived from a cross between inbred lines Qi319 and Chang7-2. QTLs for three traits, i.e. anther emergence rate (AER), anther emergence score (AES) and pollen production score (PPS) of the haploid population contributing to spontaneous chromosome doubling were evaluated in three environments. The study provides useful genetic information potentially facilitating a better understanding of the genetics of spontaneous chromosome doubling in maize as basis for efficient use in maize hybrid breeding.
Special comments
L 42: Wording: „haploids are sterile, and haploid fertility can be restored“ (see also L 57).
The sterility of haploids is a consequence of haploidy itself. Therefore fertility cannot be „restored“ in the haploid state. Instead, fertility results from doubling to reach the diploid state!
L 57: „haploid male fertility (HMF) and haploid female fertility [21],“ (see also title)
For the said reason, these terms cause confusion. Instead, the terms „paternal“ and „maternal haploid“ should be used (cf. Chalyk ST. Euphytica 1994; 79: 13-18).
L 77 „maize inbred lines Qi319 and 77 Chang7-2“(see comment on L 77)
L 325 „cross between inbred lines Qi319 and 325 Chang 7-2 [34]2“
The experimental material (parental lines) have to be described here in detail (main characteristics, origin/breeder, pedigree). Citations 34. (Li W. Analysis on in vivo haploid induction effects and machine learning application in doubled haploid technology of maize. In Beijing, China: China Agricultural University 2017. 463) and 35. (Jiao Y. The evaluation of maize haploid inducers and the study on genetic effects of F0 generation. In 464 Beijing, China: China Agricultural University 2017) are not adequate and therefore not useful because these manuscripts are not referenced and particularly not accessible for the international community. (see Materials and Methods, L 324ff)!

Author Response
Comment 1: L 42, Wording: „haploids are sterile, and haploid fertility can be restored“
Response: Thank you for your suggestion. We have corrected this description, please refer to line 44.
Comment 2: L 57, haploid male fertility (HMF) and haploid female fertility, these terms cause confusion. Instead, the terms „paternal“ and „maternal haploid“ should be used (cf. Chalyk ST. Euphytica 1994; 79: 13-18).
Response: Thank you for your comment. The haploid male fertility and haploid female fertility is generally referred to the fertility of tassel and ear of a haploid. Generally, “paternal” and “maternal” were used to describe the source of a haploid, i.e., derived from male parent or female parent. We therefore think using haploid male fertility and haploid female gametophyte fertility will be more appropriate. In material and method, we clearly described the haploids used in this research were maternal haploids induced by haploid inducer lines CAU6. Please refer to line 345-349.
Comment 3: L 77 and L 325, The experimental material (parental lines) have to be described here in detail, Citations 34 and 35 are not adequate and therefore not useful because these manuscripts are not referenced and particularly not accessible for the international community. (see Materials and Methods, L 324ff)
Response: Thank you for your valuable comment and advice. The main characteristics and breeder of inbred lines Qi319 and Chang7-2 have been added in line 341-344, The DH population used in this study was generated by Dr. Wei Li, we therefore think it is more appropriate to cite his doctoral dissertation (citation 34). While, we have corrected the description of CAU6, and deleted the citation 35. Please refer to line 345-349.
Reviewer 4 Report
The manuscript "QTL Mapping of Haploid Male Fertility Traits in Maize (Zea mays L.)" submitted to Plants has been reviewed. It was a study for QTL mapping of HMF traits. Several useful QTL were identified across maize genome which provide a basic understanding of spontaneous chromosomal doubling in maize.
The experiment design and performance were in logic, and the writing of the manuscript was pretty straightforward. I do not have many comments on it. Please see few comments and I would suggest a through proofreading before next submission.
Nice job. Thank you.

Author Response
Response: Thank you for your valuable comment and advice. We have changed “QTLs” to “QTL” all through the manuscript and added the reference on line 349. We also have corrected mistakes in the manuscript, please refer to revisions of line 20, 25, 34, 243 and 349.
Round 2
Reviewer 3 Report
- Results (L 85 ff, ref. Fig 1): The number of lines tested is obviously not tested in the paper; please add or explain.
- As already indicated before, the use of language is sometimes unusual and makes it difficult to follow. For example, the „term „DH mapping haploids“ (L 91 ff) is somehow misleading in my opinion. Wouldn’t it be better to use the term „DH-derived haploids“ instead?
- Further question: Since different haploids all represent distinct genotypes, you would have to clone each line in order to be able to test identical genotypes at the two different environments. Since this was obviously not the case here, the trials at the two places must actually represent two distinct and independent experiments with different lines (genotypes). Please elaborate!
Author Response
Response: Thank you for your suggestions and comments.
- We have changed “genetic variation” to “phenotypic variation”, please refer to line 81, and added the mean of AER, AES and PPS for DH-derived haploid population in Figure 1B. The phenotype of HMF-related traits presented a skewed distribution, so we did not calculate standard error or standard deviation. however, we did analysis of variance (Figure 1D).
- We have changed “DH mapping haploids” to “DH-derived haploids” all through the manuscript, and corrected other description where cause misleading.
- Each line of DH population had been reproduced by DH0 kernels selfed before inducing haploids. Each DH line was planted at the Hainan in 2017 and pollinated by haploid inducer line CAU6. The haploids derived from same DH line have identical genotypes, and sufficient quantity of haploids had been obtained in 2017 for further phenotype investigation. We clarified this part in section 4.1, line 279-281.